# Preoperative TAVR Planning: How to Do It

**DOI:** 10.3390/jcm11092582

**Published:** 2022-05-05

**Authors:** Rodrigo Petersen Saadi, Ana Paula Tagliari, Eduardo Keller Saadi, Marcelo Haertel Miglioranza, Carisi Anne Polanczyck

**Affiliations:** 1Post Graduate Program in Cardiology and Cardiovascular Science, Hospital de Clínicas de Porto Alegre, Federal University of Rio Grande do Sul, Porto Alegre 90410-000, Brazil; aninhatagliari@yahoo.com.br (A.P.T.); esaadi@terra.com.br (E.K.S.); carisi.anne@gmail.com (C.A.P.); 2Cardiovascular Surgery Department, Hospital São Lucas da PUC-RS, Porto Alegre 90610-001, Brazil; 3Cardiovascular Surgery Department, Hospital Mãe de Deus, Porto Alegre 90980-481, Brazil; 4EcoHaertel, Echocardiography Department, Hospital Mãe de Deus, Porto Alegre 90980-481, Brazil; marcelohaertel@gmail.com; 5Cardiology Department, Hospital Moinhos de Vento, Porto Alegre 90035-000, Brazil

**Keywords:** TAVR, sizing, planning, MDCT, 3D echocardiography, MRI

## Abstract

Transcatheter aortic valve replacement (TAVR) is a well-established treatment option for patients with severe symptomatic aortic stenosis (AS) whose procedural efficacy and safety have been continuously improving. Appropriate preprocedural planning, including aortic valve annulus measurements, transcatheter heart valve choice, and possible procedural complication anticipation is mandatory to a successful procedure. The gold standard for preoperative planning is still to perform a multi-detector computed angiotomography (MDCT), which provides all the information required. Nonetheless, 3D echocardiography and magnet resonance imaging (MRI) are great alternatives for some patients. In this article, we provide an updated comprehensive review, focusing on preoperative TAVR planning and the standard steps required to do it properly.

## 1. Introduction

Transcatheter aortic valve replacement (TAVR) has risen as a less invasive alternative for treating severe symptomatic aortic stenosis (AS) in patients at all surgical risk scores [1,2,3,4,5,6,7]. Since Alain Cribier pioneered the TAVR procedure in 2002, the advances in this field have been outstanding. Newer generation devices and different types of self and balloon-expandable valves are being released every year, and the results are remarkable improvements in procedural efficacy and safety [8].

One of the key points for a successful TAVR is carefully preprocedural planning, including accurate aortic root and aortic valve diameters measurements. Aortic angulation; aortic annulus minimum, medium and maximum diameters, area, and perimeter; left ventricle outflow tract minimum, medium and maximum diameters, area, and perimeter; sinus of Valsalva diameters; right and left coronary arteries height; sinotubular junction diameters, area, and perimeter; ascending aorta diameters; calcification distribution pattern; and C-arm angulation are all relevant information to perform a procedure properly [9].

The measurements of the aortic root and ascending aorta are used to choose the appropriate transcatheter heart valve type and size and foresee possible procedure-related complications, such as coronary artery obstruction, aortic annulus, and sinotubular junction rupture, paravalvular leak, valve embolization, and pacemaker implantation need. The size of the transcatheter heart valve that will be implanted is chosen based on the aortic annulus perimeter for the self-expandable platforms, and on the aortic annulus area for the balloon-expandable ones [10,11]. Undersizing the aortic annulus may cause paravalvular leak or valve embolization, whereas oversizing may reduce prosthesis durability, cause annulus rupture, and conduction issues leading to pacemaker implantation [12,13,14].

Initially, TAVR sizing was made using 2D echocardiography and/or during the procedure, using graduated pigtail catheters (Figure 1). However, the planning strategy evolved dramatically, especially when 3D technology started to be used since 2D echo frequently undersized the aortic annulus. The current gold standard method to perform valve measurements and preprocedural TAVR planning is the multi-detector computed angiotomography (MDCT), with an appropriate TAVR protocol [15,16]. MDCT measurements can be performed manually, using semi-automated software, or using automated software. Furthermore, the aortic annulus may be measured by magnetic resonance imaging and 3D transesophageal echocardiography. The possibility of a 3D image (MDCT or 3D echo) allowed a more precise measurement.

Herein, we provided an updated comprehensive review, focusing on preoperative TAVR planning and the standard steps required to do it properly.

## 2. Aortic Root Assessment

Understanding the aortic root anatomy and the importance of each anatomical feature involved in a TAVR procedure is fundamental to achieving a successful intervention.

### 2.1. Aortic Annulus

The aortic annulus dimension is fundamental for choosing the appropriate transcatheter heart valve type and size. The aortic annulus is defined as a virtual ring built by joining the points of the basal attachments of the aortic leaflets [17]. It is crucial to understand that the aortic annulus is a 3D structure and that 2D measures may cause mistakes. That is the reason why the gold standard for annulus evaluation is the 3D multiplanar reconstruction (MPR) of MDCT. Three-dimensional echocardiography provides similar measurements compared with MDCT [18]. On the other hand, 2D measures usually undersize the aortic annulus [19].

The aortic annulus perimeter is used to choose the size of self-expandable valves, whereas the aortic annulus area is used for the balloon-expandable ones. It is important to have the correct dimension of the annulus, to avoid oversizing or undersizing the implanted transcatheter heart valve and, therefore, avoid procedural complications (Figure 2).

### 2.2. Left Ventricle Outflow Tract (LVOT)

The LVOT is a virtual area below the mitral valve. It is a convention to measure LVOT diameters, perimeter, and area 4 mm below the virtual aortic ring. The LVOT is part of the TAVR “landing zone”, which includes the aortic annulus, aortic leaflets, and LVOT [14]. It is known that a calcified and non-tubular LVOT is associated with poor outcomes, including paravalvular leak and LVOT rupture risk [21,22]. Thus, proper analysis of the LVOT is mandatory to plan the procedure and prevent possible complications.

### 2.3. Coronary Arteries Height and Sinus of Valsalva

Coronary obstruction is a life-threatening complication following TAVR, with a mortality rate achieving up to 50%, and an incidence varying from 0.4% to 1.2% [23,24]. Low coronary ostia height and narrow sinus of Valsalva are the two main risk factors for coronary occlusion. The coronaries height is measured by tracing a straight line from the bottom of the coronary ostium until the virtual aortic annulus. The sinus of Valsalva is measured from the middle of the leaflet to the opposite commissure. Coronary arteries with a height less than 10 mm, especially if associated with small sinus of Valsalva (less than 28 mm), are associated with high coronary occlusion risk.

### 2.4. C-Arm Angulation

A perfect C-arm angulation, avoiding parallax effect, is important to have no optic illusion during the procedure. Parallax is a displacement or difference in the apparent position of an object viewed along two different lines of sight and is measured by the angle or semi-angle of inclination between those two lines. At the beginning of TAVR experience, C-arm angulation was acquired using three pigtails and a considerable degree of contrast injection to align the three aortic leaflets and prevent the aortic annulus parallax effect during TAVR deployment. Nowadays, with preoperative TAVR MDCT planning, it is possible to predict all C-arm angulations required to avoid parallax [25,26]. By unifying all these angulations, a curve is formatted, the so-called aortic valve S curve [27,28]. Before each TAVR, it is important to know the S curve for that patient to optimize the results and prevent unnecessary contrast injections. Two main angulations predictions are mandatory before TAVR: three-sinus coplanar and cusp overlap views. The three-sinus coplanar is the angulation where the three sinuses are aligned and equidistant from each other, being the non-coronary cusp at the left of the image, the right coronary cusp in the middle, and the left coronary cusp at the right. The cusp overlap view is the angulation where the non-coronary cusp is isolated at the left of the image, and the right coronary cusp is overlapping the left coronary cusp at the right part of the image.

### 2.5. Sinotubular Junction

Sinotubular junction (STJ) diameter and height are especially important for balloon-expandable valves implants since the balloon may injure the STJ in the case of a low STJ. In self-expandable supra-annular devices, valve-in-valve, and TAVR-in-TAVR, a low and narrow STJ can cause sinus sequestration with coronary malperfusion. The STJ height is measured perpendicularly to the annular plane, and the diameter is measured by the standard way [29].

### 2.6. Ascending Aorta

Assessment of any aortopathy is relevant, especially in patients with bicuspid aortic valve, when the commitment of the aorta is frequent, so measurement of the diameter of ascending aorta is part of the TAVR protocol.

### 2.7. Peripheral Access Vessels

Peripheral vessel accesses analysis, starting with femoral arteries, is relevant once most TAVR contraindications are related to inadequate accesses. If transfemoral TAVR is unsuitable, the second access of choice is trans left subclavian/axillary artery. If left subclavian/axillary is not indicated, due to inadequate diameter, calcification, or in the presence of a patent left internal mammary artery bypass, the left carotid should be analyzed to use as third transarterial alternative access. Transaortic and transapical are seldom used.

## 3. Computed Angiotomography (MDCT)

Computed angiotomography (MDCT) is the preferred method to plan a TAVR procedure by most operators [30]. The planning can be performed manually, using semi-automated and automated software. Each one has intra- and inter-observational variabilities, which will be discussed below.

### 3.1. MDCT Acquisition

The key component for well-acquired MDCT images is an ECG-synchronized MDCT that covers at least the aortic root, followed by non-ECG synchronized images of the aorta, iliac, and femoral vasculature [29]. The ECG-synchronized MDCT of the aortic root is important, because the aortic annulus undergo conformable change throughout the cardiac cycle, being bigger and circular in systole, and oval in diastole. The goal is to measure the greatest possible annular dimension, which can be found during the cardiac systole (20–40% of the cardiac cycle) [31,32].

Regarding radiation, a tube potential of 100 kV is usually indicated for patients weighing <90 kg or with a body mass index (BMI) <30 kg/m^2^, whereas a tube potential of 120 kV is indicated for patients weighing >90 kg and with BMI >30 kg/m^2^.

Intravenous contrast administration is mandatory. Optimal images require high intra-arterial opacification, and attenuation values should exceed 250 Hounsfield units. MDCT data should be reconstructed as an axial, thin-sliced multiphasic data set, with <1 mm slice thickness. Reconstruction intervals should be spaced at <10% intervals across the acquired portion of the cardiac cycle [29].

A 3D multiplanar reconstruction (MPR) of the aorta, aortic valve, and its structures is mandatory to perform TAVR planning (Figure 3).

### 3.2. Available Methods

Many different kinds of software can be used to make appropriate MDCT measurements of the aortic root, coronary ostia, and optimal angiographic deployment projections: manual, semi-automated, and automated. The manual measures are the most used by operators since they can be done by cheap or free software, such as Horos^®^ and Osirix^®.^

#### 3.2.1. Manual Sizing

The manual TAVR sizing is usually made using the 3D MPR tool of Osirix^®^ or Horos^®^ software. In the MPR mode, we have three correlated images: coronal, sagittal, and transversal. The goal is to perfectly align the virtual aortic annulus, which corresponds to the base of the three aortic cusps. The manual method does not provide information about the steps needed for sizing, and there is no automated report. In 2019, a consensus on MDCT imaging on TAVR describing the main steps was published [29]. This consensus provides further and detailed information about MDCT manual preprocedural planning. The manual measurement takes more time than the semi-automated and automated measures, and its learning curve is bigger. However, when used by experienced professionals, it may provide all the information necessary to perform a safe TAVR procedure.

There are some studies comparing the variability of measurement by different observers. These articles found a strong agreement for aortic annulus and coronary arteries height assessment for experienced observers (at least 2 years of experience) [33,34].

Furthermore, Knobloch et al. and Le Couteulx et al. reported interobserver variability in MDCTs evaluated by observers with different levels of expertise. In the Le Couteulx et al. study, Observer 1 was an expert, whereas Observer 2 was a resident physician with 6 months of practice, and Observer 3 was a trained resident physician with starting experience. Intra- and inter-observer reproducibility were excellent for all aortic annulus dimensions, with an intraclass correlation coefficient ranging, respectively, from 0.84 to 0.98 and from 0.82 to 0.97. Agreement for selection of prosthesis size was almost perfect between the two most experienced observers (*k* = 0.82) and substantial with the inexperienced observer (*k* = 0.67) [35]. In the Knobloch et al. study, Observer 1 was a radiologist with 6 years of experience, Observer 2 was a laboratory technician with 3 years of experience, and Observer 3 was a medical student with no experience. Intra-observer variability did not differ significantly. However, significant differences were found in mean inter-observer variance (*p* < 0.001). They advocate that multi-reader paradigms led to significantly increased precision compared with single readers with different levels of experience [36].

#### 3.2.2. Semi-Automated and Automated Sizing

Semi-automated software are broadly used by TAVR companies and operators around the globe. The most commonly utilized software is the 3MensioValves (3mensio Medical Imaging BV, Maastricht, The Netherlands). However, the drawback of 3MensioVales software is its high cost, preventing its broad use (Figure 4). There is another semi-automated software called ProSizeAV, which is actually a plugin to be used with Horos^®^ or Osirix^®^. However, this plugin does not have CE or FDA approval, and there are no data proving its efficacy (Figure 5).

There is another available semi-automated (syngo. viaVB20A, Siemens, Munich, Germany) software. In 2018, Horehledova et al. compared the Siemens manual and semi-automated software and demonstrated an excellent inter-software agreement (ICC = 0.93; range 0.90–0.95). The time needed for evaluation using semi-automatic assessment (3 min 24 s) was significantly lower (*p* < 0.001) compared with a fully manual approach (6 min 31 s) [37].

Lou et al. also compared manual, semi-automated, and fully automated measurement of the aortic annulus using Siemens software. Semi-automated analysis required major correction in five patients (4.5%). Mean manual annulus area was significantly smaller than fully automated results (*p* < 0.001), but similar to semi-automated measurements. The frequency of concordant recommendations for valve size increased if a manual analysis was replaced by semi-automated method (60% agreement was improved to 82.4%; 95% confidence interval for the difference [69.1–83.4%]) [38].

## 4. Echocardiography

Echocardiography is a non-invasive broadly available method used to diagnose cardiac conditions and plan cardiac procedures. The 2D transthoracic echo, at the beginning of TAVR experience, was used to size aortic annulus diameters, perimeter, and area. However, it is known that 2D echo usually underestimates the measures, thus undersizing the aortic annulus, facilitating the occurrence of paravalvular regurgitation and resulting in poor outcomes [19,39,40]. On the other hand, novel 3D transesophageal echo has been evolving and apparently, when correctly used, has a good correlation with MDCT regarding aortic annulus measures and has some advantages, such as not requiring venous contrast [18,41,42,43]. However, it is important to keep in mind that the aortic annulus measures are fundamental to choosing appropriate transcatheter heart valve size, nonetheless, there are many other important measurements as coronary arteries height, LVOT dimensions, sinus of Valsalva diameter, and ascending aorta which cannot be done properly by any 2D or 3E echo (Figure 6). Furthermore, vascular access cannot be measured by echo as well, and calcium-related artifacts may compromise echocardiography imaging.

### 3D Transesophageal Echocardiography Annulus Sizing

Elkaryoni et al. published, in 2018, a systematic review and meta-analysis about 3D TEE as an alternative to MDCT for aortic annular sizing. Thirteen studies were included (1228 patients). A strong linear correlation was found between 3D TEE and MDCT measurements of aortic annulus area (r = 0.84, *p* < 0.001), mean perimeter (r = 0.85, *p* < 0.001), and mean diameter (r = 0.80, *p* < 0.001). They concluded the 3D TEE demonstrated a high level of correlation with those evaluated by MDCT, and that 3D TEE is a feasible choice for aortic annulus assessment, with advantages of real-time assessment, lack of contrast, and no radiation exposure [44]. Another systematic review and meta-analysis comparing 3D TEE and MDCT sizing was published by Mork et al., in 2021. In this paper, a total of 889 patients from ten studies were included. Pooled correlation coefficients between 3D TEE and MDCT of annulus area, perimeter, area derived-diameter, perimeter derived-diameter, maximum and minimum diameter measurements were strong 0.89 (95% CI: 0.84–0.92), 0.88 (95% CI: 0.83–0.92), 0.87 (95% CI: 0.77–0.93), 0.87 (95% CI: 0.77–0.93), 0.79 (95% CI: 0.64–0.87), and 0.75 (95% CI: 0.61–0.84) (overall *p* < 0.0001), respectively [45].

In another systematic review and meta-analysis, Rong et al. also reported strong correlation between 3D TEE and MDCT annular area, annular perimeter, annular diameter, and left ventricular outflow tract area measurements (0.86 [95% CI, 0.80–0.90]; 0.89 [CI, 0.82–0.93]; 0.80 [CI, 0.70–0.87]; and 0.78 [CI, 0.61–0.88], respectively) [46].

On the other hand, Vaquerizo et al. reported a single-center cohort study comparing 3D TEE and MDCT, and stated that 3D TEE-derived measurements were significantly smaller compared with MSCT: perimeter (68.6 + 5.9 vs. 75.1 + 5.7 mm, respectively; *p* < 0.0001); area (345.6 + 64.5 vs. 426.9 + 68.9 mm^2^, respectively; *p* < 0.0001). The percentage difference between 3D TEE and MSCT measurements was around 9%. Agreement between MSCT- and 3D TEE-based THV sizing (perimeter) occurred in 44% of patients. Using the 3D TEE perimeter annular measurements, up to 50% of patients would have received an inappropriate valve size according to manufacturer-recommended, area-derived sizing algorithms [47].

Similarly, Singh et al. reported 185 patients between 2013 and 2015 and stated that the undersize of echo sizing may reduce even patients’ survival. 2D and 3D TEE underestimated the annulus size by −1.49 and −1.32, respectively, and discrepancies >10% between TEE and MDCT were associated with a decrease in post-implant survival [48].

Aortic short axis (upper esophageal, 40–45°), and long axis (mid esophageal 120–140°), are the most used probe positions to evaluate the aortic valve annulus.

## 5. Magnetic Resonance Imaging

The use of magnetic resonance imaging (MRI) for TAVR planning has been increasing in the last years. Although 3D TEE is an alternative for the aortic annulus sizing for patients who cannot undergo MDCT due to contrast allergy or renal failure, it is not possible to perform all fundamental measures to plan the entire procedure through echo evaluation, and MRI has emerged as a feasible alternative to MDCT [49]. However, MRI is significantly inferior to MDCT defining the presence and extension of valvular and vascular calcium, which is an important feature for TAVR.

In 2016, Ruile et al. compared MDCT with non-contrast MRI for aortic root assessment, and the agreement for hypothetical prosthesis sizing was found in 63 of 67 (94%) of patients. However, accesses were not evaluated in this study [50].

Mayr et al. performed a pilot study in 16 patients comparing MDCT with a dedicated MRI protocol including non-contrast 3D “whole heart” acquisition and contrast-enhanced 3D aortoiliofemoral MRI, and MRI demonstrated a very strong correlation (r = 0.956, *p* < 0.0001) and complete consistency between MRI and MDCT regarding the decision for valve size. Vessel luminal diameters and angulation of aortoiliofemoral access also showed very strong correlation (r = 0.819 to 0.996, *p* < 0.001) [51]. A total non-contrast MRI protocol for TAVI guidance was developed by the same group. A comparison between MDCT and non-contrast MRI in 26 patients demonstrated a moderate to strong correlation for assessment of minimal vessel diameter for aortoiliofemoral access (r = 0.572 for the right external iliac artery to r = 0.851 for the thoracic descending aorta, *p* = 0.002 and *p* ≤ 0.0001, respectively), with good-to-excellent inter-observer reliability (ICC 0.862 to 0.999, all *p* < 0.0001), whereas mean diameters of the infrarenal aorta and iliofemoral vessels differed significantly (bias 0.37 to 0.98 mm, *p* = 0.041 to <0.0001) (Figure 7) [52].

As MRI requires reliable compensation strategies to deal with cardiac and respiratory motion artifacts including ECG triggering and respiratory navigator gating, Pammiger et al. compared a simpler self-navigated with a navigator-gated non-contrast 3D whole-heart MRI and found high to very high correlation for aortic root measurements (*p* < 0.0001), concluding that a self-navigator is feasible and achieve similar results to navigator-gated MRI, with shorter acquisition time [53].

Similarly, Aouad et al. validated a faster single breath-hold MRI acquisition (k-t acceleration to 3D cine b-SSFP MRI) [54].

## 6. Discussion

Preoperative TAVR imaging planning is fundamental to achieve procedural success. MDCT with 3D MPR remains the gold standard for aortic root and iliofemoral system evaluation. Semi-automated software (3MensioValves and Siemens) play an important role since they have a low inter- and intra-observer variability, whereas the ProSizeAV plugin still demands CE and FDA approval. The drawback of these semi-automated software is still their high cost. Manual measurements provide accurate results when performed by experienced professionals, although they have a longer learning curve compared with semi-automatic software. Totally automated software still need approval to become the gold standard method, since they usually overestimate aortic annulus measures. Furthermore, MDCT provides a full assessment of the aortic root (aortic annulus, LVOT, and sinus of Valsalva dimensions, coronary arteries height, STJ, and ascending aorta evaluation), besides iliofemoral or alternative accesses (axillary/subclavian, carotid, direct aortic, transapical and transcaval) evaluation.

Although 2D images, such as 2D echocardiography, underestimate aortic annulus dimensions, 3D TEE has proved to be an excellent alternative to MDCT to perform aortic annulus sizing and appropriate transcatheter heart valve selection. The major advantage of 3D echo is that it is a minimally invasive exam, which does not need contrast injection or radiation. However, although it is feasible to measure aortic annulus and LVOT dimensions with 3D echo, usually it is not possible to properly size coronary arteries height, and it is not precise for MDCT to measure sinus of Valsalva and ascending aorta diameters. Furthermore, the access vessels cannot be evaluated as well.

Magnetic resonance imaging has arisen as an alternative for patients who cannot receive iodine intravenous contrast. The correlation between MDCT and MRI for aortic root dimensions is excellent. However, for peripheral accesses evaluation, historically, gadolinium contrast used to be administered. Today, it is possible to evaluate the access with MRI without gadolinium administration, nonetheless, the vessels’ diameters do not match perfectly. The disadvantages of MRI are its high cost, more time to acquire the image, and less experience of the operators to perform the measurements. This imaging method will certainly increase in the next years.

## 7. Conclusions

Careful preprocedural planning is mandatory to achieve a successful TAVR procedure and avoid serious complications. There are many ways to acquire and evaluate the anatomical details required to perform a safe TAVR planning (aortic annulus, LVOT, sinus of Valsalva, coronary arteries, STJ, ascending aorta, and access sites evaluation) including 3D echo, MDCT, and MRI. Although MDCT is still the gold standard, it is important to be familiarized with alternative methods and know the pros and cons of each one, in order to choose the most appropriate method or a combination of them for each specific patient.

## Figures and Tables

**Figure 1 jcm-11-02582-f001:**
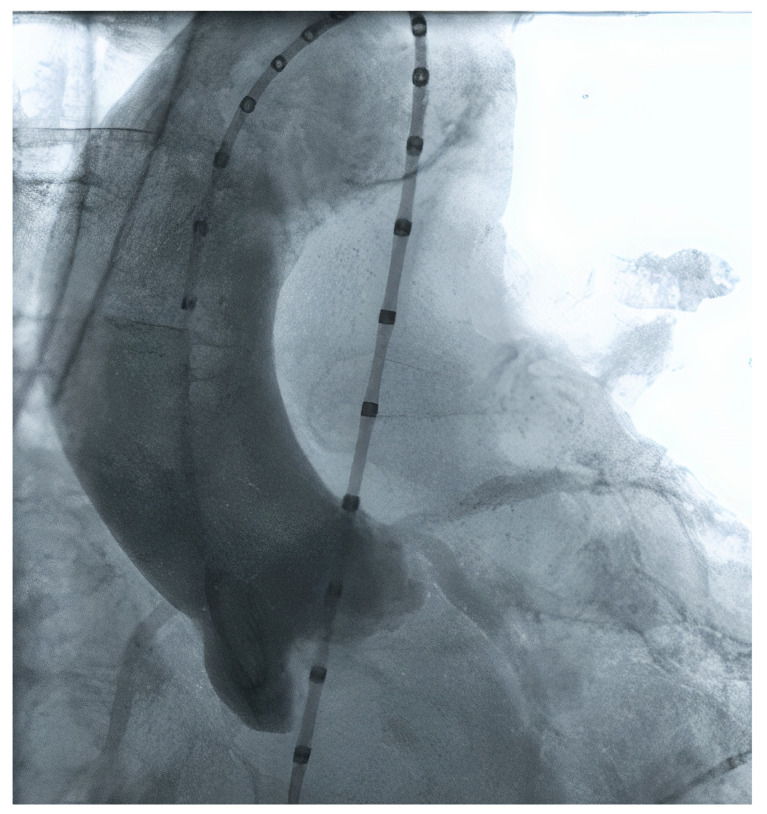
Aortic valve measurement using contrast injection from a pigtail catheter.

**Figure 2 jcm-11-02582-f002:**
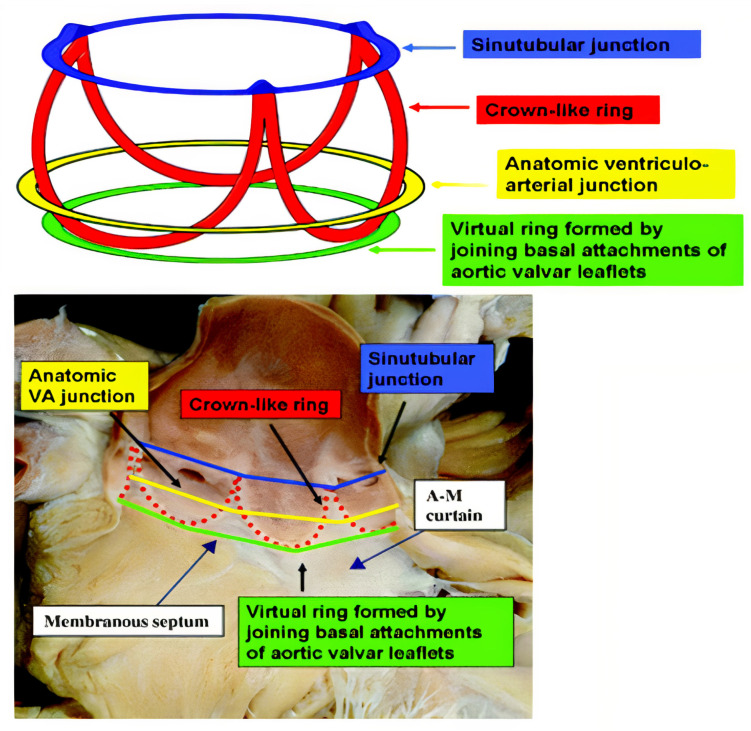
Virtual aortic annulus, sinotubular junction, and coronary arteries anatomy (adapted from Zarayelyan A. et al. [20]).

**Figure 3 jcm-11-02582-f003:**
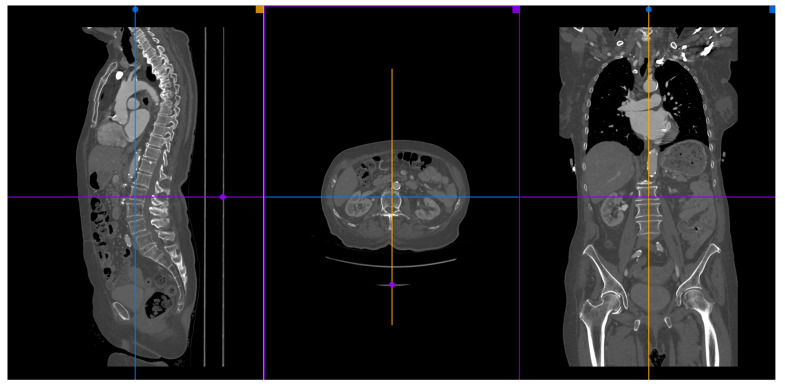
A MPR reconstruction from MDCT images using the Horos^®^ software.

**Figure 4 jcm-11-02582-f004:**
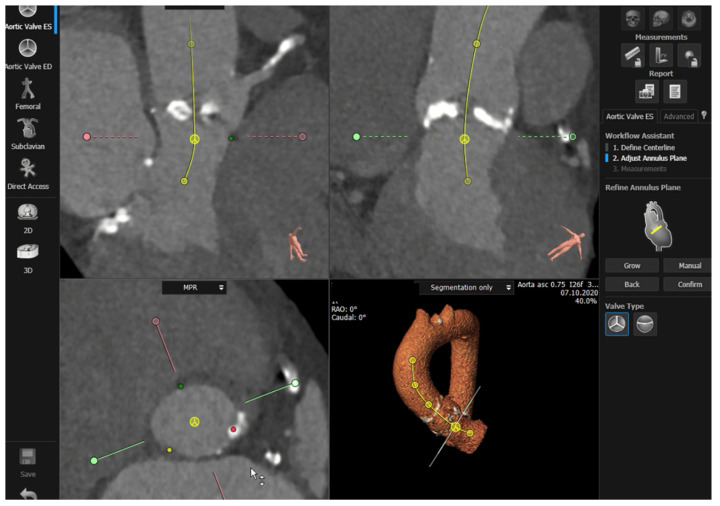
Measurements performed using the 3MensioValves.

**Figure 5 jcm-11-02582-f005:**
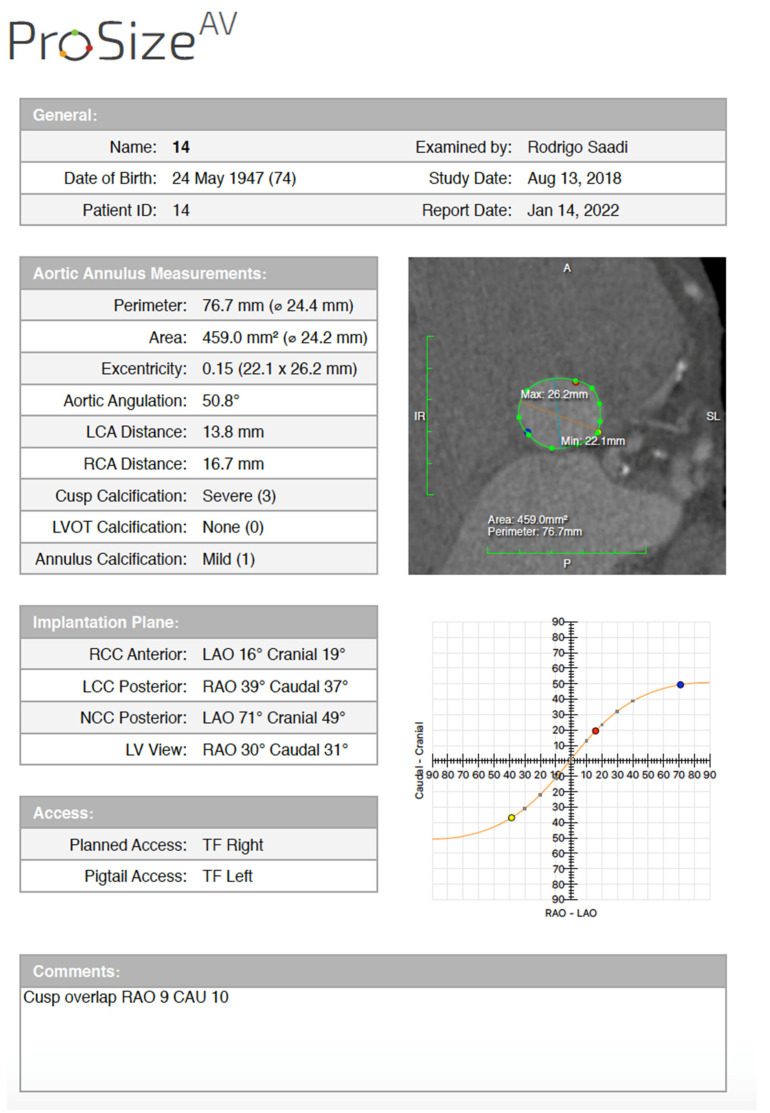
ProSizeAV report.

**Figure 6 jcm-11-02582-f006:**
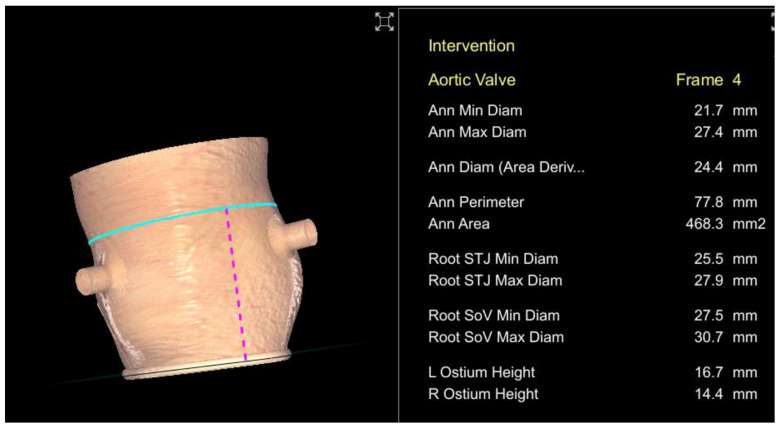
Example of a 3D transesophageal echocardiography aortic root assessment.

**Figure 7 jcm-11-02582-f007:**
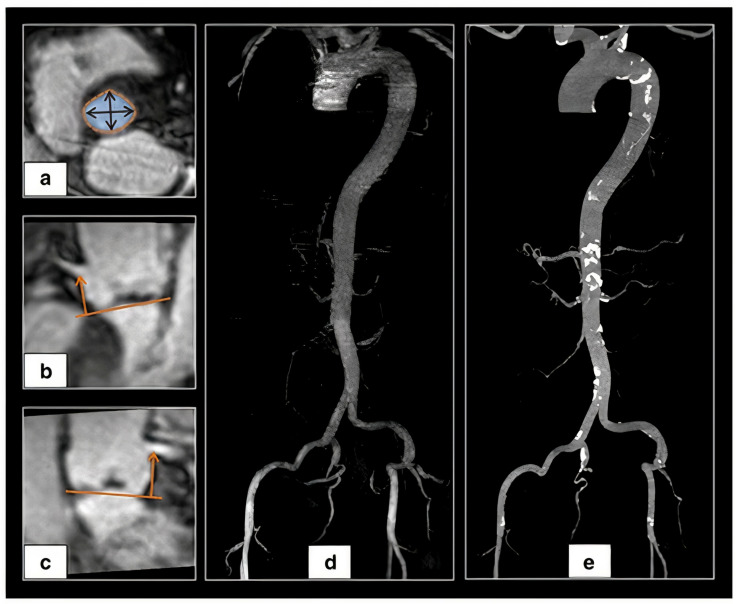
Example of a non-contrast 3D ‘whole heart’ MRI (**a**) aortic annular, arrows showing the minimum and maximum diameters, (**b**) right coronary and (**c**) left coronary arteries height (orange arrows), (**d**) maximum intensity projection of aortoiliofemoral MRI and (**e**) MDCT image (adapted from Pammiger et al. [52]).

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
