# Peer review of "Preoperative TAVR Planning: How to Do It"

_jcm, 2022, doi:10.3390/jcm11092582_

Round 1

Reviewer 1 Report

The work described in this paper is of current interest, the paper is well structured and the picture material is fine. Fig 1 could be better in terms of contrast. Language needs a minor revisions due to style and grammar. references are appropriate. Since decisions making prior to valve implantation is always the crucial part the current work contributes to the related field and adds knowledge.

Author Response

The authors really appreciate your comments.

We have modified figure 1 according to your suggestion and reviewed the entire article in order to improve language mistakes.

Thank you very much for your time.

Kind regards,

The Authors

Reviewer 2 Report

The manuscript entitled "Preoperative TAVR planning: how to do it" shows a nice and practical review according to imaging in previous TAVR. 

Some comments should be done: 

- Page 1, line 43: consider changing the expression "possible procedure catastrophes" to "possible procedural related complications". 
- C-arm angulation: consider including a mention about which angulations should be included in the report (3-sinus coplanar and cusp overlap).
- Page 5, line166: reference 29 should be mentioned as the main reference in this field. Consider including a paragraph inviting readers to look for it for further and detailed information about CT preprocedural planning. Reference 33 is old-fashioned and it should be deleted.
- Semi-automated and automated sizing: the authors mention some commercial software (Osirix, Horos, ProSize, 3Mensio, Siemens); but they don't mention other available software such as GE, Philips, FEOPS, etc. To avoid any possible commercial bias, I would recommend including all the software available or not including the specific name of anyone in the main text. It would be referenced.
- Page 8, line 224: "which can not be measured properly by echo", neither the vascular access. Please, mention it in the manuscript. 
- TOE: There is no mention of the calcium-related artifacts or possible transesophageal probe positions to improve AV annulus definition. 
- MRI: The MRI is significantly inferior to defining the presence and extension of valvular and vascular calcium, an aspect crucial for TAVR. This significant bias should be mentioned in the manuscript.
- References are not according to Vancouver normative
- Figure 7: adapted from Mayr A et al. It should be mentioned if specific permission was obtained prior to reproducing this image.

Author Response

The authors really appreciate your comments. We are certain that your great comments have contributed a lot to improving the article.

1 - We have changed the expression "possible procedure catastrophes" to "possible procedural related complications", as suggested.
2 - We added the two most important angulations (3-sinus coplanar and cusp overlap views), as suggested.
3 - We excluded reference 33 and keep reference 29 as the main reference in the field. Furthermore, we added a phrase inviting the readers to read reference 29 to further information, as suggested. 
4 - The softwares mentioned are all softwares to use MDCT as the main image to perform the pre-operative TAVR planning. I think GE and FEOPS are both echocardiography softwares. The authors don't know any other MDCT softwares besides these mentioned in the text. 
5 - It was added to the text that vascular access can not be properly measured by echo, as suggested.
6 - It was added to the text the possible calcium-related artifacts and the main probe position used to aortic annulus measurement by TOE, as suggested. 
7 - It was added to the text the fact that MRI is significantly inferior to defining the presence and extension of valvular and vascular calcium, as suggested.
8 - References were corrected to fit the journal needs, as suggested.
9 - Figure 7 was changed and permission was already obtained, as suggested. 

Thank you very much for your time.

Kind Regards,

The Authors

Reviewer 3 Report

I read with great interest and attention the paper entitled "Preoperative TAVR planning: how to do it"
As a vascular surgeon strongly involved in the procedures of EVAR, TEVAR, and F / B-EVAR, I am strongly aware of how much correct sizing and planning is fundamental for the success (immediate and remote) of an aortic surgery.
However, I have not honestly understood what the authors have attempted to do with the present paper.
To me the manuscript seems like a sort of vademecum for planning, but without the research elements typical of a scientific paper.                         Perhaps this text would be more suitable for a cardiology fellows textbook.
Where are the novelty elements and the research of the typical sources of a scientific review? What do the authors want to prove?

Author Response

The authors would like to acknowledge the reviewer's comments. The main purpose of this article review is to provide an up-to-date review focusing on the aortic valve measurements needed to perform a safe TAVI procedure. As a review article, we collected the information available from previous reliable and prestigious sources, bringing it all together and facilitating the search for information by implanters. We really believe that this type of information fits very well in this special edition focused on transcatheter procedures and will be very useful for quick sources before any procedural planning.

Thank you

Kind regards, 

The authors

Reviewer 4 Report

Dear Authors,

Thank you for the work done with excellence and quality.

More and more practical and scientifically based needs to be able to carry out our surgical procedures with more precision and better defining the situation of our patients with heart disease.

Increasingly, the transcatheter world is becoming a global reality. Defining, planning and carrying out are steps with extreme importance in the result and in the quality of the result mainly.

In this review, we find the step by step for a TAVR and consequently comparisons in the exams for planning for use and/or not.

Very interesting the way they put it with pros and cons.

Please correct only the words that are spelled wrong.

Excellent images.

Bests Regards,

Author Response

The authors really appreciate your kind comments.

We have reviewed the text and corrected some words that were spelled wrong, as suggested.

We would like to thank you very much for your time.

Kind regards,

The Authors

Round 2

Reviewer 3 Report

Despite the review carried out by the authors, I have to confirm my previous idea. I still cannot understand what the authors have tried to do with this article. The manuscript appears to be a sort of vademecum for the design of the TAVR, but without the research elements typical of a scientific paper. I remain convinced that a text like this will be more suitable for a cardiology textbook.

Author Response

We would like to thank you a lot for your comments, but again we understand that this review article with pre-operative information is indeed necessary. It is so important that the other three reviewers have already approved and made many complements to the article, and the Editors of this Special Edition understand that review articles are needed to supplement the theme of this Special Edition.

Thanks again,

Kins regards,

The authors